# Predictors of Prostate Cancer at Fusion Biopsy: The Role of Positive Family History, Hypertension, Diabetes, and Body Mass Index

Marco Oderda *,† , Alessandro Dematteis † , Giorgio Calleris, Adriana Conti, Daniele D'Agate, Marco Falcone, Alessandro Marquis, Gabriele Montefusco, Giancarlo Marra and Paolo Gontero

Division of Urology, Department of Surgical Sciences, Molinette Hospital, University of Turin, 10126 Turin, Italy
* Correspondence: marco.oderda@unito.it; Tel.: +39-0116707682
† These authors contributed equally to this work.

**Abstract:** Background: PSA density and an elevated PI-RADS score are among the strongest predictors of prostate cancer (PCa) in a fusion biopsy. Positive family history, hypertension, diabetes, and obesity have also been associated with the risk of developing PCa. We aim to identify predictors of the prostate cancer detection rate (CDR) in a series of patients undergoing a fusion biopsy. Methods: We retrospectively evaluated 736 consecutive patients who underwent an elastic fusion biopsy from 2020 to 2022. Targeted biopsies (2–4 cores per MRI target) were followed by systematic mapping (10–12 cores). Clinically significant PCa (csPCa) was defined as ISUP score ≥ 2. Uni- and multivariable logistic regression analyses were performed to identify predictors of CDR among age, body mass index (BMI), hypertension, diabetes, positive family history, PSA, a positive digital rectal examination (DRE), PSA density ≥ 0.15, previous negative biopsy status, PI-RADS score, and size of MRI lesion. Results: The median patients' age was 71 years, and median PSA was 6.6 ng/mL. A total of 20% of patients had a positive digital rectal examination. Suspicious lesions in mpMRI were scored as 3, 4, and 5 in 14.9%, 55.0%, and 17.5% of cases, respectively. The CDR was 63.2% for all cancers and 58.7% for csPCa. Only age (OR 1.04, $p < 0.001$), a positive DRE (OR 1.75, $p = 0.04$), PSA density (OR 2.68, $p < 0.001$), and elevated PI-RADS score (OR 4.02, $p = 0.003$) were significant predictors of the CDR in the multivariable analysis for overall PCa. The same associations were found for csPCa. The size of an MRI lesion was associated with the CDR only in uni-variable analysis (OR 1.07, $p < 0.001$). BMI, hypertension, diabetes, and a positive family history were not predictors of PCa. Conclusions: In a series of patients selected for a fusion biopsy, positive family history, hypertension, diabetes, or BMI are not predictors of PCa detection. PSA-density and PI-RADS score are confirmed to be strong predictors of the CDR.

**Keywords:** fusion biopsy; prostate cancer; risk factors; family history; hypertension; diabetes; body mass index

## 1. Introduction

Prostate cancer (PCa) is the second most common tumor diagnosed in men, accounting for an estimated 1.4 million diagnoses globally in 2020 [1,2]. Some not modifiable risk factors for overall PCa incidence have been demonstrated and consist in older age, a family history of PCa, and ancestry, in particular, the African American race [3–5]. PCa incidence and mortality change according to patients' geographical origin. In particular, this heterogeneity can be attributed to a different intensity in PSA screening among different populations. However, some research on people migration points out a possible role of environmental and lifestyle factors in disease risk and etiology [6–9]. Some of these modifiable risk factors such as obesity, hypertension, smoking, and dietary habits have been associated with fatal prostate cancer and with the progression of the disease [10–12]. Multiparametric magnetic resonance imaging (mpMRI) of the prostate is now recommended by European

Association of Urology (EAU) guidelines before a prostate biopsy [13]. In the era of mpMRI of the prostate as the first diagnostic step when PCa is suspected, elevated PI-RADS scores and PSA-density are among the strongest predictors of PCa in a fusion biopsy. Our aim is to identify and evaluate possible predictors of the prostate cancer detection rate (CDR) in a monocentric series of patients with a radiological suspicion of PCa undergoing an mpMRI-guided biopsy.

## 2. Materials and Methods

This was a retrospective study that evaluated the data of 736 consecutive patients who underwent a transrectal (TRUS)–mpMRI elastic fusion biopsy with the Koelis Trinity™ system (Koelis, Meylan, France) between 2020 and 2022, under local anesthesia in an outpatient setting. All patients had at least one region defined as suspicious for cancer in mpMRI and underwent targeted biopsies (2–4 cores per mpMRI target), followed by systematic mapping (10–12 cores), as previously described [14]. Exclusion criteria were as follows: (1) PSA > 20 ng/mL; (2) age > 80; (3) previous PCa diagnosis; (4) colostomy or rectal amputation; (5) congenital coagulation alterations and/or those who did not interrupt anticoagulant therapy (aspirin was permitted to undergo the biopsy; patients using anticoagulants or antiplatelet agents other than aspirin discontinued treatment or bridged with heparin depending on the risk deriving from therapy discontinuation); (6) no antibiotic prophylaxis; (7) no consent for study participation; and (8) insufficient follow-up information. Further, mpMRIs were performed in different centers, reflecting current clinical practice. Suspicious lesions were scored according to the PI-RADS classification, version 2 [15]. Clinically significant PCa (csPCa) was defined as ISUP score $\geq$ 2. All patients signed informed consent for the use of clinical information for clinical studies (coordinator ethics committee protocol number 0040478). The study was performed according to the Standard for Reporting Diagnostic Accuracy Studies (STARD) [16].

For this current study, a positive family history was defined as $\geq$2 first- or second-degree relatives with PCa on the same side of the pedigree; hypertension was defined as (1) repeated blood pressure measurements above 140/90 mmHg, according to the European Society of Cardiology (ESC) guidelines 2018, or (2) being under medical treatment for hypertension; (mellitus) diabetes was defined (1) according to the ESC guidelines 2019 or (2) as being under medical treatment for diabetes; and Body Mass Index (BMI) was defined as weight in kg divided by the square of height in meters [kg/m$^2$] at the time of the biopsy. The prostate volume was reported on the MRI report in the majority of the cases or, if missing, was calculated using the ellipsoid formula (prostate length $\times$ width $\times$ height $\times$ 0.52).

Statistical analyses were performed with SPSS version 28.0 (IBM Corp, Armonk, NY, USA). Quantitative data are shown as mean and standard deviation (SD), or median and interquartile range (IQR) and were compared using the Mann–Whitney test, whereas qualitative data are shown as frequencies and percentages and were compared using Pearson's chi-squared test. Uni- and multi-variable logistic regression analyses were performed to identify predictors of the CDR among age, body mass index (BMI), hypertension, diabetes, a positive family history, PSA, a positive digital rectal examination (DRE), PSA-density $\geq$ 0.15, previous negative biopsy status, PI-RADS score, and size of mpMRI lesion. Missing data were treated with pairwise deletion. Statistical significance was considered at 2-sided $p < 0.05$.

## 3. Results

The patients' baseline characteristics are detailed in Table 1. The median patients' age was 71 years (IQR 65–76); median PSA was 6.6 ng/mL (IQR 5.0–9.5); median PSA-d was 0.14 (IQR 0.09–0.21); and median prostate volume was 48.5 cc (IQR 35–69). A positive digital rectal examination (DRE), performed by the urologist in charge of the biopsy just before the procedure, was reported in 20% of patients. A total of 168 patients (22.8%) had 1 or more previously negative prostate biopsies. A single mpMRI target was identified in 73.9% of cases, with a mean diameter of 11.5 mm (SD 6.2). The remaining 26.1% of cases harbored 2 targets in mpMRI. Suspicious lesions in mpMRI were scored as 3, 4, and

5 in 14.9%, 55.0%, and 17.5% of cases, respectively. The CDR was 63.2% for all cancers and 58.7% for csPCa, with most cancers being diagnosed as ISUP 2 (42.8%) and 3 (31.0%). When comparing patients with positive and negative prostate biopsy results, statistically significant differences were found in age, PSA density, positive DRE, prostate volume, and target area characteristics (number, size, and PI-RADS score). These differences are graphically presented in Figure 1.

**Table 1.** Patients' characteristics.

| Variables | All Patients | Missing Data | Patients with Positive Biopsy | Patients with Negative Biopsy | *p* |
|---|---|---|---|---|---|
| **Patients** | 736 | - | 465 (63.2%) | 271 (36.8%) | |
| **Age; years; median (IQR)** | 71 (11) | 1 (0.1%) | 72 (11) | 69 (10) | <0.001 |
| **BMI; mean (SD)** | 25.8 (3.4) | 370 (50.2%) | 25.8 (3.4) | 25.9 (3.5) | 0.78 |
| **Hypertension; *n* (%)** | 399 (54.2%) | 13 (1.8%) | 253 (63.4%) | 146 (36.6%) | 1.00 |
| **Diabetes; *n* (%)** | 66 (9%) | 27 (3.7%) | 46 (69.7%) | 20 (30.3%) | 0.28 |
| **Positive family history for PCa; *n* (%)** | 55 (7.5%) | 97 (13.2%) | 36 (65.5%) | 19 (34.5%) | 0.77 |
| **PSA; ng/mL; median (IQR)** | 6.5 (4.3) | 7 (0.9%) | 6.8 (4.5) | 6.1 (3.8) | 0.13 |
| **PSA density; ng/mL/mL; median (IQR)** | 0.14 (0.12) | 170 (23.1%) | 0.16 (0.12) | 0.11 (0.10) | <0.001 |
| **PSA density $\geq$ 0.15; *n* (%)** | 261 (35.5%) | 170 (23.1%) | 198 (75.9%) | 63 (24.1%) | <0.001 |
| **Positive DRE; *n* (%)** | 150 (20.4) | 37 (5.0%) | 112 (74.7%) | 38 (25.3%) | 0.002 |
| **Prostate volume; cc; median (IQR)** | 48 (35) | 168 (22.8%) | 42 (26) | 60 (40) | <0.001 |
| **Previous negative biopsies; *n* (%)** | 168 (22.8%) | 5 (0.7%) | 105 (62.5%) | 63 (37.5%) | 0.85 |
| **Single mpMRI target; *n* (%)** | 544 (73.9%) | 55 (7.5%) | 340 (62.5%) | 204 (37.5%) | 0.009 |
| **Size of targets; mm; mean (SD)** | 11.5 (6.2) | 106 (14.4%) | 12.3 (6.9) | 10.1 (4.4) | <0.001 |
| **PIRADS of targets (*maximum score if multiple*); *n* (%)** | | | | | |
| -     3 | 110 (14.9%) | | 45 (40.9%) | 65 (59.1%) | |
| -     4 | 405 (55.0%) | 92 (12.5%) | 271 (66.9%) | 134 (33.1%) | <0.001 |
| -     5 | 129 (17.5%) | | 109 (84.5%) | 20 (15.5%) | |
| **Cancer detection rate; *n* (%)** | 465 (63.2%) | 0 (0%) | 465 (63.2%) | - | - |
| **Clinically significant cancer detection rate; *n* (%)** | 432 (58.7%) | 0 (0%) | 432 (58.7%) | - | - |
| **PCa ISUP score; *n* (%)** | | | | | |
| -     1 | 33 (7.0%) | | 33 (7.0%) | | |
| -     2 | 199 (42.8%) | | 199 (42.8%) | | |
| -     3 | 144 (31.0%) | | 144 (31.0%) | | |
| -     4 | 61 (13.2%) | | 61 (13.2%) | | |
| -     5 | 28 (6.0%) | | 28 (6.0%) | | |

Uni- and multi-variable analyses are shown in Table 2. The significant predictors of overall PCa detection in multivariate analysis were age (OR 1.04, 95% CI 1.02–1.07, $p < 0.001$), PSA density (OR 2.68, 95% CI 1.73–4.15, $p < 0.001$), and elevated PI-RADS score (OR 4.02, 95% CI 1.62–9.96, $p = 0.003$). As for csPCa, significant predictors in multivariate analysis were age (OR 1.04, 95% CI 1.01–1.07, $p < 0.001$), positive DRE (OR 1.75, 95% CI 1.01–3.02, $p = 0.04$), PSA density (OR 2.47, 95% CI 1.62–3.76, $p < 0.001$), and elevated PI-RADS score (OR 3.56, 95% CI 1.50–8.45, $p = 0.004$). The size of an mpMRI lesion was associated with the CDR only in uni-variable analysis, for both overall PCa (OR 1.07, 95% CI 1.03–1.10, $p < 0.001$) and csPCa (OR 1.05, 95% CI 1.02–1.08, $p < 0.001$). Posi-

tive family history, hypertension, diabetes, and BMI were not found to be predictors of PCa detection.

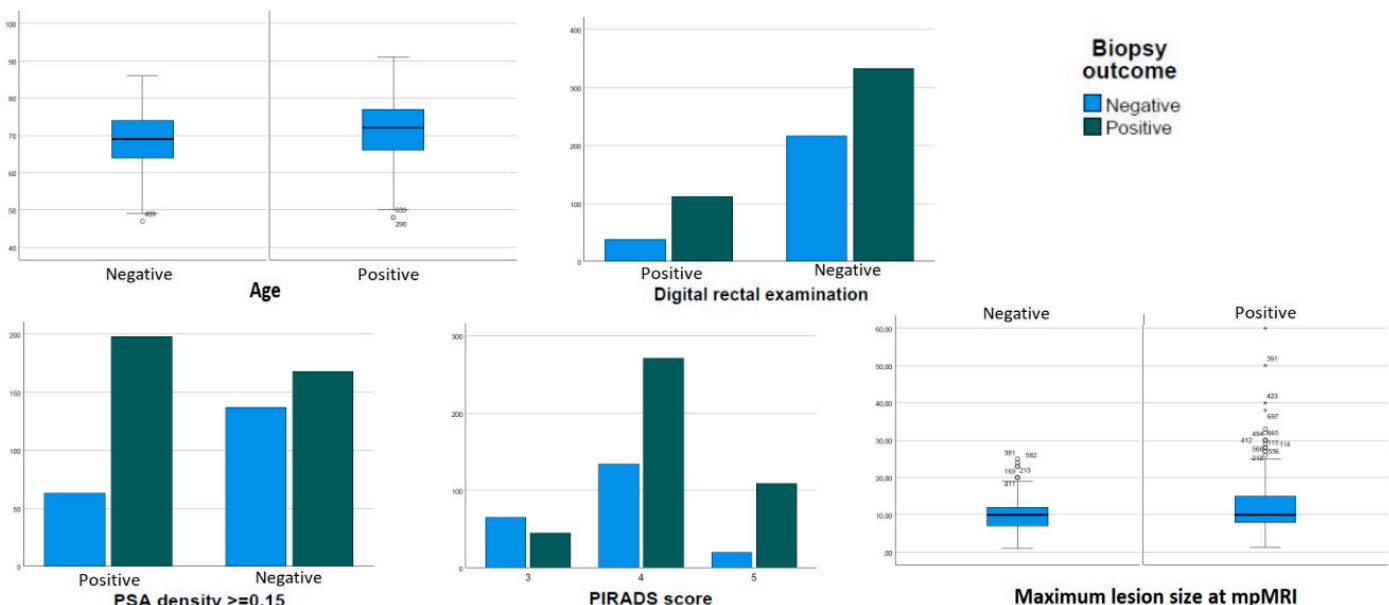

**Figure 1.** Predictors of PCa and biopsy outcome.

**Table 2.** Predictors of PCa and clinically significant PCa at uni- and multi-variable analysis.

| Variable | All PCa | | CsPCa | |
|---|---|---|---|---|
| | Uni-Variable | Multi-Variable | Uni-Variable | Multi-Variable |
| Age | **1.04 (1.02–1.06)** | **1.04 (1.02–1.07)** | **1.05 (1.03–1.07)** | **1.04 (1.01–1.07)** |
| | *p* < 0.001 | *p* < 0.001 | *p* < 0.001 | *p* < 0.001 |
| Body mass index | 0.99 (0.93–1.05) | - | 0.97 (0.91–1.03) | - |
| | *p* = 0.78 | | *p* = 0.33 | |
| Hypertension | 1.00 (0.74–1.36) | - | 0.98 (0.73–1.33) | - |
| | *p* = 0.97 | | *p* = 0.93 | |
| Diabetes | 1.38 (0.80–2.40) | - | 1.36 (0.80–2.31) | - |
| | *p* = 0.24 | | *p* = 0.25 | |
| Positive family history for PCa | 1.12 (0.63–2.01) | - | 1.20 (0.68–2.12) | - |
| | *p* = 0.68 | | *p* = 0.52 | |
| PSA (ng/mL) | 1.02 (0.99–1.05) | - | 1.01 (0.99–1.04) | - |
| | *p* = 0.15 | | *p* = 0.22 | |
| Positive DRE | **1.91 (1.27–2.86)** | 1.47 (0.84–2.59) | **2.15 (1.45–3.20)** | **1.75 (1.01–3.02)** |
| | *p* = 0.002 | *p* = 0.17 | *p* < 0.001 | *p* = 0.04 |
| PSA density ≥ 0.15 | **2.56 (1.78–3.68)** | **2.68 (1.73–4.15)** | **2.41 (1.70–3.42)** | **2.47 (1.62–3.76)** |
| | *p* < 0.001 | *p* < 0.001 | *p* < 0.001 | *p* < 0.001 |
| Previous negative biopsy | 0.95 (0.66–1.36) | - | 0.91 (0.64–1.29) | - |
| | *p* = 0.79 | | *p* = 0.61 | |

**Table 2.** *Cont.*

| | All PCa | | CsPCa | |
|---|---|---|---|---|
| **Variable** | **Uni-Variable** | **Multi-Variable** | **Uni-Variable** | **Multi-Variable** |
| PIRADS score | - | - | - | - |
| 3 | 2.92 (1.89–4.50) | 2.74 (1.61–4.68) | 3.31 (2.19–5.17) | 3.31 (1.91–5.73) |
| 4 | *p* < 0.001 | *p* < 0.001 | *p* < 0.001 | *p* < 0.001 |
| | 7.87 (4.27–14.48) | 4.02 (1.62–9.96) | 7.81 (4.35–14.02) | 3.56 (1.50–8.45) |
| 5 | *p* < 0.001 | *p* = 0.003 | *p* < 0.001 | *p* = 0.004 |
| Size of the lesion (mm) | 1.07 (1.03–1.10) | 1.03 (0.98–1.09) | 1.05 (1.02–1.08) | 1.04 (0.98–1.10) |
| | *p* < 0.001 | *p* = 0.19 | *p* < 0.001 | *p* = 0.12 |

## 4. Discussion

Some risk factors for overall PCa incidence have been proved, and they include older age, ethnicity, and a positive family history of this disease. These are all non-modifiable risk factors. Older age is strongly linked to the risk of PCa, and patients over 60 years have an odds ratio for PCa diagnosis of 5.35 (95% CI 2.17–13.19) in multi-variable models [17] The diagnosis of this neoplasm is rare before the age of 40, whereas it increases seriously after 55 years of age [10]. The average age patients are diagnosed is 66 years old [18]. Incidence and mortality change dramatically depending on race and ethnicity worldwide. In the US, the incidence among black men is approximately 60% higher than white men, and PCa is responsible for a 2.4-times greater mortality rate for African American compared to Caucasian patients. Asians/Pacific Islanders, America Indians/Alaskan Natives, and Hispanic men have a lower PCa incidence and mortality rates compared to non-Hispanic white men [10,19]. There is some evidence that this variation in PCa rates is due to genetic factors [20], differences in diagnostic practices and treatment, and patients' lifestyle [9,21]. Several studies established a familial aggregation of PCa diagnosis: An up to 3-times increased risk of PCa has been demonstrated among men with a first-degree relative with a positive history of this neoplasm and nearly 9-fold higher for men with both a father and brother affected [22]. Most of the familial aggregation of PCa derives from shared genetic factors, as observed in studies conducted on twins [23]. More than 60% of early onset of PCa cases are characterized by a positive family history of this tumor, and more than 40% had a first degree affected relative [24].

A different association has been observed, analyzing several risk factors for indolent compared to lethal disease [25]. Hence, it is mandatory to distinguish between risk factors for overall PCa and those for aggressive disease. Many modifiable risk factors for advanced disease have been studied. One of them is obesity, which is defined as a BMI $\geq$ 30 kg/m$^2$. The relationship between neoplasm incidence and body size is complicated; however, recent studies suggest that obesity is linked to a higher risk of PCa-specific mortality and recurrence [26]. Obesity has been associated with an almost six-times higher risk of being affected by PCa in some retrospective studies (OR 5.79, 95% CI: 2.66–12.57) [27]. On the contrary, a sub-analysis of the REDUCE trial, in which a prostate biopsy was conducted at fixed timepoints, has found no differences for obese patients in terms of overall PCa incidence, yet an increased risk in terms of clinically-significant disease in multi-variable analysis (OR 1.28, 95% CI 1.01–1.63) [11]. Moreover, overweightness has been associated with an increased cancer-specific mortality in a large meta-analysis including 280199 patients (HR: 1.19, 95% CI 1.10–1.28), showing a dose-response relationship [28]. Several biological mechanisms have been invoked to explain this association in the obese patient: (1) an alteration of the insulin/insulin-like grow factor (IGF)-1 signaling promoting mitogenesis and pro-angiogenesis and inhibiting apoptosis, (2) a decreased blood sex hormones concentration (testosterone and dihydrotestosterone), potentially leading to poorly differentiated cancer, (3) an imbalance in adipocyte-derived hormones both at a systemic and paracrine level (increased leptin and decreased adiponectin), promoting inflammation and

angiogenic factors, and (4) microcirculation alterations leading to hypoxia, oxidative stress, and the development of reactive oxygen radicals species [29].

Hypertension has been studied as a possible modifiable risk factor; however, research investigating its role in PCa etiology is very poor. However, Esposito et al. compiled 10 studies (in total 4343 PCa cases) revealing a positive association between hypertension and PCa risk (a significant 15% greater risk, $p = 0.035$) [12]. This correlation could be explained by an elevated sympathetic nervous system activity that can lead to an androgen mediated stimulation of tumoral cells [30]. In the literature, there is no positive significant association between hyperglycemia or type 2 diabetes and prostate cancer risk [12,31]. When these conditions are studied as a whole, under the definition of metabolic syndrome, the literature does not report unanimous results, describing both a negative and positive association with PCa and even no relation at all. Examining good-quality data from a prostate biopsy cohort of 2235 patients, no single metabolic syndrome component showed an association with PCa; however, the risk of overall and clinically significant PCa increased with the number of metabolic syndrome components [32].

Possible explanations for the heterogeneity of the impact of metabolic risk factors for PCa incidence and aggressiveness reported in the literature involve (1) a lower prevalence of metabolic syndrome among people with higher income and education, who might have an easier access to screening and diagnosis programs in some healthcare systems [33] and (2) differences in baseline PCa risk of study populations and in the definitions adopted.

In the last years, mpMRI has become an essential exam in the diagnostic process of PCa. An mpMRI-guided fusion biopsy has significantly higher detection rates of clinically significant PCa than an ultrasound-guided standard biopsy alone, especially in a repeated biopsy setting [34]. In our previous research, a software-guided elastic fusion biopsy has achieved good targeting precision and reproducibility in daily clinical practice [35]. Analyzing these results, we questioned the role of traditional risk factors for PCa in a homogeneous series of patients selected for a fusion biopsy. In other words, we inquired whether a patient with a suspicious lesion in mpMRI has a further increased risk of having PCa if he has other risk factors, or not.

In this article, we present the results of a single center, retrospective study collecting the data of 736 consecutive patients undergoing a fusion biopsy (performed by highly experienced urologists, with more than 500 cases each), where we evaluated the role of risk factors including age, hypertension, diabetes, BMI, and a positive family history. The diagnostic work-up, from the initial clinical suspicion of prostate cancer until the execution of mpMRI, was carried out in different institutions, sometimes independent from ours, reflecting everyday clinical practice and repeatable scenarios.

Several findings in this study deserve attention. First, in our series of patients we noticed a CDR of 63.2% for all cancers and 58.7% for csPCa, witnessing a good accuracy in line with the results of the PRECISION trial [36]. Second, older age was positively associated with the PCa detection rate, confirming the strength of this risk factor and once more the fact that PCa is a disease in elderly men. The only other significant variables were PSA density, which is now a well-known risk factor for PCa when PSA-d $\geq 0.15$ [37], and an elevated PI-RADS score, keeping in mind that a PI-RADS 5 in mpMRI is by far the strongest predictor of PCa. Third, in our homogeneous series of patients with suspicions of PCa in mpMRI, a positive familial history, hypertension, and elevated BMI seemed to lose their role as predictors of PCa. Therefore, it would seem that when there is a clinical suspicion of prostate cancer confirmed by a lesion in mpMRI, there is no other particular risk factor in the medical history that can further increase the probability of detecting a PCa with a fusion biopsy. We were unable to investigate the ethnicity/race role as a risk factor, due to the homogeneity of the patient population included. We acknowledge the retrospective nature and the relatively small sample size as the main limitations of our study.

## 5. Conclusions

In a series of patients with suspicions of PCa in mpMRI undergoing a prostate fusion biopsy, the presence of an elevated PI-RADS score remains the major predictor of the CDR, and older age and PSA-density are confirmed to be strong predictors of PCa. On the other hand, a traditional risk factor such as a positive family history does not act as a predictor of PCa. Hypertension, diabetes, or an elevated BMI do not show any association with PCa detection. To our knowledge, this is the first work to analyze predictors of PCa in a series of patients already selected for an mpMRI-guided biopsy. Among the limitations of this study, we acknowledge the retrospective design and the limited sample size.

**Author Contributions:** Conceptualization, M.O. and A.D.; methodology, G.M. (Giancarlo Marra); software, A.M.; validation, D.D. and G.M. (Gabriele Montefusco); formal analysis, M.F.; investigation, G.C.; resources, P.G.; data curation, M.O.; data collection, A.C. and A.D.; writing—original draft preparation, A.D. and M.O.; writing—review and editing, P.G., M.O. and G.M. (Giancarlo Marra); visualization, M.O.; supervision, P.G.; project administration, M.O. All authors have read and agreed to the published version of the manuscript.

**Funding:** This research received no external funding.

**Institutional Review Board Statement:** The study was conducted in accordance with the Declaration of Helsinki and applicable ethical standards. Ethical approval was waived for this study due to observational retrospective nature of the study. All patients signed an informed consent for the procedure, including data processing for institutional research activity.

**Informed Consent Statement:** Informed consent was obtained from all subjects involved in the study.

**Data Availability Statement:** The data presented in this study are available on motivated request from the corresponding author, for patient privacy reasons.

**Conflicts of Interest:** Marco Oderda has worked as consultant for Koelis™.

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
