# Peer review of "Predictors of Prostate Cancer at Fusion Biopsy: The Role of Positive Family History, Hypertension, Diabetes, and Body Mass Index"

_curroncol, doi:10.3390/curroncol30050374_

Round 1

Reviewer 1 Report

- Literature review is poor and needs to extend.

- Materials and methods is very short.

- Resolution of figure 1 is not sufficient.

- Conclusion section is necessary.

_ Native English person need to improve the whole of the manuscript.

Author Response

- Literature review is poor and needs to extend. Thank you for this suggestion. We extended the discussion section reporting more evidence from the literature, although there is scarcity of consistent data to be reported
- Materials and methods is very short. Material and methods were revised adding more information about the performance of the study.
- Resolution of figure 1 is not sufficient. We uploaded figure 1 as TIF file, which should guarantee an acceptable quality for publication. 
- Conclusion section is necessary. Thank you for this suggestion. We added the Conclusion section as follows: “In a series of patients with suspicion of PCa at mpMRI undergoing prostate fusion biopsy, the presence of an elevated PI-RADS score remains the major predictor of CDR, and older age and PSA-density are confirmed to be strong predictors of PCa. On the other hand, a traditional risk factor such as positive family history does not act as a predictor of PCa. Hypertension, diabetes, or BMI no not show any association with PCa. To our knowledge, this is the first work to analyze predictors of PCa in a series of patients already selected for a mpMRI-guided biopsy. Among the limitations of the study, we acknowledge the retrospective design and the limited sample size.”
- Native English person need to improve the whole of the manuscript. The manuscript was revised by a English native speaker.

Reviewer 2 Report

the authors are presenting interesting study regarding risk factors for prostate cancer diagnosis in a cohort of patients, which undergo prostate MRI/US fusion biopsy - an everyday clinical scenario for all urologists

In this reviewer`s opinion, there are several issues that should be taken into account by the authors

Abstract - CDR should be clarified - abbreviation used before explanation in Introduction

Introduction - Good and concise paragraph

Material and Methods - good description of the methodology of the study, nicely depicting its protocol.

Results 

- raw 76-77 positive DRE - at what point it has been performed and by whom? 

- raw 78-79 - a percentage is given for the patients with solitary mpMRI lesion. What about the other 26,1 % of patients - two or more lesions?

- tabl 1. - clear and highly illustrative formatting.

BMI data missing in over half of the patients? - making it hardly suitable for sound scientific analysis

PSA volume and hence PSAD - also significant number of missing data (approximately 1/4) - although not so serious as in BMI, needs an explanation - PSA volume is readily available both through MRI and TRUS

missing ISUP in 12,5% ? that needs detailed explanation?

Discussion - good and clear paragraph joining authors` results with current literature. 

A dedicated Conclusions paragraph would be beneficial for the manuscript quality - with information on raw 161-167 - in this reviewer`s opinion this is the essence of the study

Some general remarks - references are outdated - especially regarding epidemiology of PCa - should be totally revised  

Regarding all the aforementioned, my recommendation is to accept this manuscript for publication after satisfactory comments and appropriate modification by the authors on the abovementioned issues. 

Author Response

The authors are presenting interesting study regarding risk factors for prostate cancer diagnosis in a cohort of patients, which undergo prostate MRI/US fusion biopsy - an everyday clinical scenario for all urologists. In this reviewer`s opinion, there are several issues that should be taken into account by the authors

Abstract - CDR should be clarified - abbreviation used before explanation in Introduction. Thank you. We clarified in the abstract “prostate cancer detection rate (CDR)”.

Introduction - Good and concise paragraph. We thank the reviewer for this comment.

Material and Methods - good description of the methodology of the study, nicely depicting its protocol. We thank the reviewer for this comment.

Results 

- raw 76-77 positive DRE - at what point it has been performed and by whom? Thank you for the comment. We clarified this point as follows: “A positive digital rectal examination (DRE), performed by the urologist in charge of the biopsy just before the procedure, was reported in 20% of patients.”

- raw 78-79 - a percentage is given for the patients with solitary mpMRI lesion. What about the other 26,1 % of patients - two or more lesions? Correct. We added the following sentence: “The remaining 26.1% of cases harbored two targets at mpMRI.”

- tabl 1. - clear and highly illustrative formatting.

BMI data missing in over half of the patients? - making it hardly suitable for sound scientific analysis

PSA volume and hence PSAD - also significant number of missing data (approximately 1/4) - although not so serious as in BMI, needs an explanation - PSA volume is readily available both through MRI and TRUS

missing ISUP in 12,5% ? that needs detailed explanation?

We acknowledge that missing data represent a major limitation of this retrospective study. Unfortunately, we were not able to retrieve all the data concerning BMI nor PSA density. As for ISUP, we actually have all the data, we apologize for this mistake in data reporting. We corrected the table.

Discussion - good and clear paragraph joining authors` results with current literature. We thank the reviewer for this comment.

A dedicated Conclusions paragraph would be beneficial for the manuscript quality - with information on raw 161-167 - in this reviewer`s opinion this is the essence of the study. We thank the reviewer for this suggestion: we added a dedicated Conclusion paragraph as follows: “In a series of patients with suspicion of PCa at mpMRI undergoing prostate fusion biopsy, the presence of an elevated PI-RADS score remains the major predictor of CDR, and older age and PSA-density are confirmed to be strong predictors of PCa. On the other hand, a traditional risk factor such as positive family history does not act as a predictor of PCa. Hypertension, diabetes, or BMI no not show any association with PCa. To our knowledge, this is the first work to analyze predictors of PCa in a series of patients already selected for a mpMRI-guided biopsy. Among the limitations of the study, we acknowledge the retrospective design and the limited sample size.”

Some general remarks - references are outdated - especially regarding epidemiology of PCa - should be totally revised  

We thank the reviewer for this remark: we added several new references as suggested.

Regarding all the aforementioned, my recommendation is to accept this manuscript for publication after satisfactory comments and appropriate modification by the authors on the abovementioned issues. 

Reviewer 3 Report

To date, MRI-targeted biopsy represent the standard approach for the diagnosis of clinically significant prostate cancer.  

In the literature, various studies have investigated factors predicting cancer detection rates.

In the present paper, the authors sought to identify and evaluate the predictors of cancer detection rate in a single-center series of patients with a radiological suspicion of PCa undergoing mpMRI-guided biopsy.

Overall, the paper is well-written.

Title: accurate

Abstract: reflects the report

Introduction:  clearly states background

M&M: please provide more specific definition for prostate biopsy indication: was the need based on PSA level, PSA density, other biomarkers and/or suspicious DRE and/or imaging (i.e., PI-RADS >3 or ≥3) ?

Results:  well described. Tables and figures summarize all data appropriately.

Discussion: the authors review related literature and place the findings of this work within the field.

Conclusions: the interpretation is clear and warranted

Author Response

To date, MRI-targeted biopsy represent the standard approach for the diagnosis of clinically significant prostate cancer.  

In the literature, various studies have investigated factors predicting cancer detection rates.

In the present paper, the authors sought to identify and evaluate the predictors of cancer detection rate in a single-center series of patients with a radiological suspicion of PCa undergoing mpMRI-guided biopsy. Overall, the paper is well-written.

We thank the reviewer for this comment.

Round 2

Reviewer 2 Report

The authors have made sufficient changes and comments according the reviewers` remarks. 

It is this reviewer`s opinion the manuscript to be accepted in present form